# Clinico-Radiological Phenotype of *UBTF* c.628G>A Pathogenic Variant-Related Neurodegeneration in Childhood: A Case Report and Literature Review

**DOI:** 10.3390/brainsci12091262

**Published:** 2022-09-17

**Authors:** Ching-Shiang Chi, Hsiu-Fen Lee, Chi-Ren Tsai

**Affiliations:** 1Division of Pediatric Neurology, Children’s Medical Center, Taichung Veterans General Hospital, 1650, Taiwan Boulevard Sec. 4, Taichung 407, Taiwan; 2Department of Post-Baccalaureate Medicine, College of Medicine, National Chung Hsing University, 250, Kuo Kuang Rd., Taichung 402, Taiwan

**Keywords:** childhood onset, neurodegeneration, neuroimage, *UBTF* gene

## Abstract

Background: This work aims to describe the clinico-radiological phenotype of *UBTF* c.628G>A (p.Glu210Lys) pathogenic variant-related neurodegeneration in childhood. Methods: We describe the progress of clinical and neuroimaging features in a male individual who had childhood-onset neuroregression and carried the heterozygous *UBTF* c.628G>A (p.Glu210Lys) pathogenic variant. Clinical cases reported in the literature are reviewed. Results: Fifteen individuals, from 14 reported cases and the index case, were noted. The median age at onset of neurodegeneration was 3 years. Clinical phenotype was consistent among the affected individuals, with progressive motor, speech, cognitive, and social–emotional regression together with ataxia and prominent pyramidal and extrapyramidal symptoms and signs in early to middle childhood. All individuals had the same brain MRI features in terms of symmetric and diffuse T2 high signal intensity over the bilateral subcortical, periventricular, and peritrigonal white matter and progressive cortical and subcortical supratentorial atrophy. Two individuals were reported to have bilateral thalamic involvement. All individuals had profound intellectual disability with loss of verbal and/or ambulatory functions during follow-up. Conclusions: Individuals with the heterozygous *UBTF* c.628G>A (p.Glu210Lys) pathogenic variant had consistent clinical progress and neuroimaging features. Familiarity with this clinico-radiological phenotype may allow earlier diagnosis of this rare disease.

## 1. Introduction

Neurodegenerative disorders in childhood comprise a large, heterogeneous group of diseases that share a core set of features, including regression and progressive deterioration of neurologic function with a gradual loss of previously acquired motor, sensory, cognitive, speech, vision, hearing, or locomotion functions, often associated with seizures. Genetically determined neurodegenerative disorders can be divided either clinically, according to the age of presentation and involved brain regions on neuroimaging features, or pathophysiologically, based on the involved biochemical pathways and cellular organelles [1,2]. These diseases often pose a great challenge to clinicians in terms of diagnosis and management. Advances in genomic medicine have begun to unravel the underlying causative genes of this disease entity and furthermore to provide a better understanding of the molecular pathogenesis of newly recognized rare diseases.

A new childhood-onset neurodegenerative disease affecting ribosomal metabolism has been shown to be associated with a monogenic *UBTF* (Upstream Binding Transcription Factor) (NM_014233.3) c.628G>A (p.Glu210Lys) gene variant [3,4,5,6,7]. *UBTF* is a protein coding gene for UBF (Upstream Binding Factor) that not only plays a critical role in ribosomal RNA transcription as a key component of the pre-initiation complex, mediating the recruitment of RNA polymerase I to ribosomal DNA promoter regions, but also plays important roles in chromatin remodeling and pre-ribosomal RNA processing [3,4]. The monoallelic *UBTF* c.628G>A (p.Glu210Lys) gene variant confers a gain of function to UBF with the production of a markedly increased amount of ribosomal RNA and alteration of nucleoli size and number, leading to defects in ribosomal DNA chromatin status and aberrant ribosomal RNA metabolism and ribosome biogenesis [3].

To date, 14 individuals carrying de novo heterozygous *UBTF* c.628G>A (p.Glu210Lys) pathogenic variant-related neuroregression in childhood have been reported [3,4,5,6,7]. Herein, we report a further case and review the clinico-radiological phenotype of this rare disease.

## 2. Materials and Methods

### 2.1. Case Report 

This male individual, aged 14 years and 3 months, was the first child of healthy nonconsanguineous Taiwanese parents; one younger sibling was healthy. He was born after an uneventful pregnancy, and the birth parameters were normal. Family history showed that his paternal cousin had epilepsy and autistic behavior.

The patient was referred at 7 years of age for neurodevelopmental regression. He achieved normal developmental milestones in the first year of life: he could roll over at 5 months, was able to sit alone at 6 months, was crawling at 8 months, was capable of pulling an object to stand up at 10 months (gross motor), could approach an object with pincer grasp at 10 months (fine motor), and was able to express babbles at 12 months (speech). At the age of 1 year 6 months, mild developmental delay was found by his parents: he could walk alone for 4 to 5 steps, but he could not make a mark on paper or say “papa” and “mama”. An early intervention program was started. The initial evidence of neurological regression at 2 years of age was noted by the parents who found that their son was no longer able to produce speech. At the age of 4 years, he showed an apathetic face, walked slowly, and stopped playing with other children. He also had emotional difficulties, such as a lack of motivation, had no interests at all, and had no apparent feeling toward things. At the age of 4 years 8 months, brain magnetic resonance imaging (MRI) showed T2-weighted imaging (T2WI) high signal intensity over the bilateral anterior, dorsomedial, and pulvinar nuclei of the thalamus and the bilateral periventricular and peritrigonal white matter and mild generalized cortical atrophy (Figure 1a–c). At the age of 5 years, he exhibited wide-based and waddling gait. Easy fatigue after daily activity was noticed. At the age of 7 years, motor apraxia and postural instability became obvious.

At presentation, physical examination revealed that his head circumference was 48 cm (less than 3rd percentile), body weight was 19 kg (3rd to 15th percentile), body height was 112 cm (3rd percentile), and that he had no organomegaly and no dysmorphic features. Neurological examination showed intact cranial nerves, generalized hypertonicity with left side predominance, brisk deep tendon reflexes (DTRs), positive Babinski sign, negative ankle clonus, and ataxic gait with left toe walking and left pes valgus. The other developmental domains were globally delayed. He showed poor eye contact and pincer grasp when reaching things and no speech.

Extensive workup including ammonia, lactate, assays of urinary organic acids, tandem mass spectrometry, plasma amino acids, and cerebrospinal fluid amino acids, acylcarnitine profile, lysosomal enzymatic activities, array-based Comparative Genomic Hybridization, and mitochondrial gene analysis were unremarkable. At the age of 7 years 2 months, follow-up brain MRI showed bilateral extensive thalamic lesions, with progressively diffuse, symmetric, and obvious T2WI high signal intensity over the bilateral subcortical, periventricular, and peritrigonal white matter, and progressive cortical and subcortical supratentorial atrophy (Figure 1d–f).

Whole-genome sequencing was then performed on Illumina NovaSeq 6000. Parental written informed consent was obtained prior to the collection of a blood sample for molecular studies (Figure 2). The result revealed de novo heterozygous UBTF (NM_014233.3) c.628G>A (p.Glu210Lys) pathogenic variant, which was confirmed by Sanger sequencing. The pathogenicity prediction score of UBTF (NM_014233.4) c.628G>A (p.Glu210Lys) for SIFT was 0.021 (Damaging); Polyphen2_HVAR was 0.986 (Probably damaging); MutationTaster was 1 (Disease causing); and CADD was 25.8. The allele frequency of UBTF c.628G>A (p.Glu210Lys) was 0 in gnomAD Genomes and was 0 in gnomAD Exomes. No additional pathogenic/likely pathogenic gene variants were found. This study was approved by the Institutional Review Board of Taichung Veterans General Hospital (TCVGH IRB CE20022A).

Thereafter, he displayed a downhill course with progressive deterioration. At the age of 8 years 6 months, he was unable to walk unassisted and showed cardinal features of parkinsonism, including no facial expression, start hesitation, small steppage gait, and postural instability while walking. Reduced pain sensation following a fall was observed. By the age of 10 years, he had an epileptic seizure, which was described as eye staring, eye deviation to the right side, and yelling, followed by generalized tonic–clonic seizures. At the age of 11 years, the electroencephalography (EEG) showed generalized background slowing and continuous rhythmic 3 to 4 Hz slow waves on the left frontal, temporal, and occipital regions (Figure 3a). From the age of 12 to 13 years, two interictal EEGs showed a pattern of rhythmic delta wave on the left hemisphere that became diffuse at the age of 13 years (Figure 3b,c). At age of 13 years 11 months, brain MRI revealed progressive changes over the whole thalamus and the aforementioned lesions accompanied by evident cerebellar atrophy (Figure 1g–i). At the age of death at 14 years and 3 months, he was wheelchair-bound, nonverbal, had profound intellectual disability, had difficulty swallowing, and exhibited brisk DTRs with generalized spasticity and rigidity, as well as dystonia.

### 2.2. Literature Review

We searched the PubMed database using the keyword “*UBTF*” and reviewed articles that had clinical case reports [3,4,5,6,7]. Individuals with detailed medical records, including clinical manifestations, neuroimaging features, and clinical outcomes were included in the analysis.

## 3. Results

Fifteen individuals, 6 males and 9 females, from 14 reported cases and the index case were diagnosed with heterozygous *UBTF* (NM_014233.3) c.628G>A (p.Glu210Lys) pathogenic variant-related neurodegeneration in childhood [3,4,5,6,7]. The clinical and neuroimaging features are described in detail in Appendix A.

As summarized in Table 1, prior to neurodegeneration, 10 of 15 individuals exhibited developmental delay at younger than or equal to the age of 2 years. The median age at onset of neurodegeneration was 3 years, ranging from 2 to 7 years old. Seven individuals exhibited initial developmental regression in both motor and speech skills, six subjects first showed regression in motor skill, and in two subjects, the initial observed deficit was in speech skill. Eight individuals had borderline or acquired microcephaly at follow-up. Regarding disease progression, six individuals had epilepsy, and the median age of epilepsy onset was 10.5 years, ranging from 5 to 15 years old. The seizure types could be focal and/or generalized epilepsies and the electroencephalographic findings were unremarkable or epileptiform discharges.

Based on the available data reported in the literature, all individuals showed positive pyramidal signs such as brisk DTRs and spasticity and variable degrees of extrapyramidal signs, including 11 having dystonia, 3 with chorea, and 3 with parkinsonism. Ten individuals exhibited ataxic gait. All individuals had the same brain MRI features in terms of diffuse T2 high signal intensity over the bilateral subcortical, periventricular, and peri-trigonal white matter, and progressive cortical and subcortical supratentorial atrophy; 11 showed cerebellar atrophy; and 2 had bilateral thalamic involvement. All individuals had profound intellectual disability with loss of verbal and/or ambulatory functions.

## 4. Discussion

Neurodegenerative disorders in children pose a unique diagnostic challenge. Unlike many genetic syndromes, the clinical manifestations of childhood neurodegenerative diseases are often nonspecific and show considerable overlap. Pathognomonic clinical signs are rare. These children often present in the early stages of their illness when evidence of progression is questionable and motor or cognitive impairment is relatively mild. It is only with extended observation that both clinical and neuroimaging abnormalities evolve to suggest a limited set of diagnostic possibilities.

The whole clinical picture of *UBTF* (NM_014233.3) c.628G>A (p.Glu210Lys) pathogenic variant-related childhood onset neurodegeneration is consistent among affected individuals. With or without developmental delay before the age of 2 years, motor and speech skills are the first to decline. Subsequently, social and emotional skills regress in early to middle childhood, with the clinical manifestations of apathetic face, quiet parallel playing with other children, and a lack of motivation, as described in the index case. Behavioral disorders including hyperactivity, impulsivity, and compulsive and repetitive behaviors have also been reported [4]. As the disease progresses, gait instability (frequent falls and moving slowly) with reduced sensation to noxious stimuli, ataxia, pyramidal tract signs (hyperreflexia and spasticity), and then extrapyramidal tract signs (rigidity, dystonia, chorea, and parkinsonism), mixed with or without epilepsy, develop. Eventually, loss of all function domains accompanies profound intellectual disability with non-verbal and/or non-ambulatory status.

Clinical symptoms and signs of this disease entity are nonspecific as other neurodegenerative disorders have similar features. In such circumstances, neuroimaging plays an essential role in the diagnostic approach. Affected subjects had a similar radiological phenotype in terms of symmetric and diffuse T2WI high signal intensity over the bilateral subcortical, periventricular and peritrigonal deep white matter and progressive supratentorial cortical and subcortical atrophy [3,4,5,6,7]. White matter signal was isointense to the cortical gray matter signal, indicating axonal loss and demyelinating representing Wallerian degeneration [4]. As shown in Figure 1, in cases where sequential neuroimages were available, cortical atrophy was noted initially and appeared to precede the development of cerebellar atrophy [3]; in addition, thinning of the corpus callosum was seen and progressed in severity with increasing age [4].

It is worth noting that the index case showed bilateral thalamic involvement from the anterior, dorsomedial, and pulvinar nuclei extending to the whole thalamus. Further investigation of the correlation of clinical manifestations and connectivity in the cortico-basal ganglia-thalamo-cortical circuit of this rare childhood-onset degenerative disorder is needed to obtain potentially clinically useful disease biomarkers. In addition, bilateral thalamic lesions in neurodegenerative disorders are uncommon. The differential diagnoses include lysosomal storage disorders, respiratory chain disorders, thiamine metabolism dysfunction syndromes, Wilson disease, or pyruvate dehydrogenase complex deficiency [8,9,10]; these disorders can be further narrowed by evaluating the patient’s history, imaging characteristics, and the presence or absence of lesions outside the bilateral thalami.

Taken together, the sequences of clinical features and neuroimaging findings of bilateral thalamic involvement from the anterior, dorsomedial, and pulvinar nuclei of the thalamus to whole thalamus and the diffuse, symmetric T2WI high signal intensity over the bilateral subcortical, periventricular, and peritrigonal white matter constitute the characteristic clinico-radiological phenotype of *UBTF* c.628G>A (p.Glu210Lys) pathogenic variant-related childhood onset neurodegeneration. Clinical and neuroimaging information may provide an earlier clue when diagnosing this rare disease, which may prompt genetic testing to obtain an accurate diagnosis.

## 5. Conclusions

This paper presents further evidence related to the disease-predisposing allele, c.628G>A (p.Glu210Lys) in *UBTF*, and further describes the clinical features of early-onset neurodevelopmental delay and regression as well as the neuroimaging findings of progressive bilateral thalamic lesions, diffuse and symmetric abnormal signal intensity over periventricular and peritrigonal deep white matter, and progressive cortical and subcortical atrophy. Familiarity with the evolutionary changes of clinical and radiological phenotypes of this rare neurodegenerative disorder may be helpful when diagnosing this disease.

## Figures and Tables

**Figure 1 brainsci-12-01262-f001:**
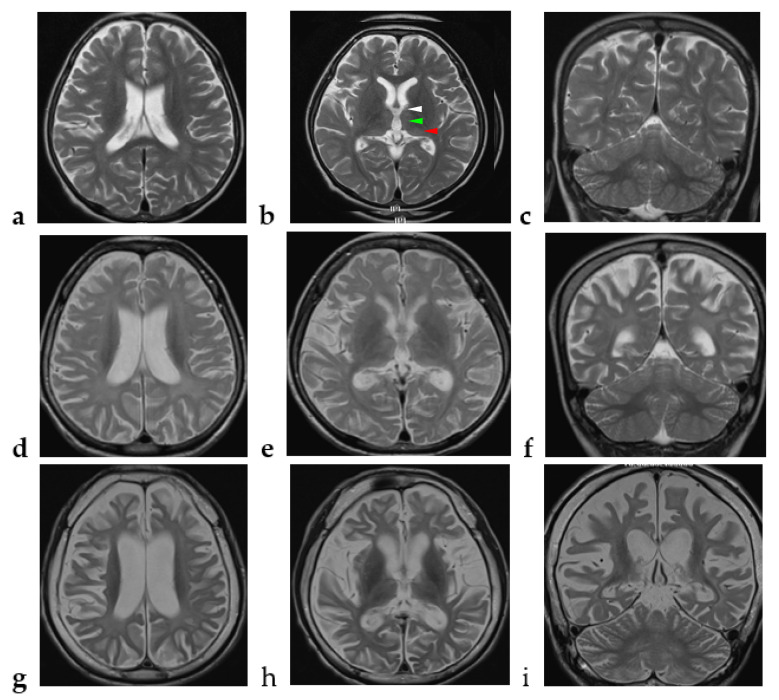
Chronological changes of the brain MRI in the index case. (**a**–**c**) At the age of 4 years and 8 months, axial view (TE 106 ms/ TR 6300 ms) and coronal view (TE 105 ms/ TR 5433 ms) of brain MRI show high signal intensity over the bilateral anterior (white arrow head), dorsomedial (green arrow head), and pulvinar (red arrow head) nuclei of the thalamus and the bilateral periventricular and peritrigonal white matter and generalized cortical atrophy. (**d**–**f**) At the age of 7 years and 2 months, axial view (TE 84 ms/ TR 4500 ms) and coronal view (TE 91 ms/ TR 4500 ms) of brain MRI show extensive high signal intensity over the bilateral thalamus; diffuse, symmetric, and obvious high signal intensity over the bilateral periventricular and peritrigonal white matter; and progressive cortical and subcortical supratentorial atrophy. (**g**–**i**) At the age of 13 years and 11 months, axial view (TE 102 ms/ TR 3430 ms) and coronal view (TE 83 ms/ TR 3530 ms) of brain MRI show progressive changes over the whole thalamus and the aforementioned areas and evident cerebellar atrophy with high signal intensity over the bilateral cerebellar white matter.

**Figure 2 brainsci-12-01262-f002:**
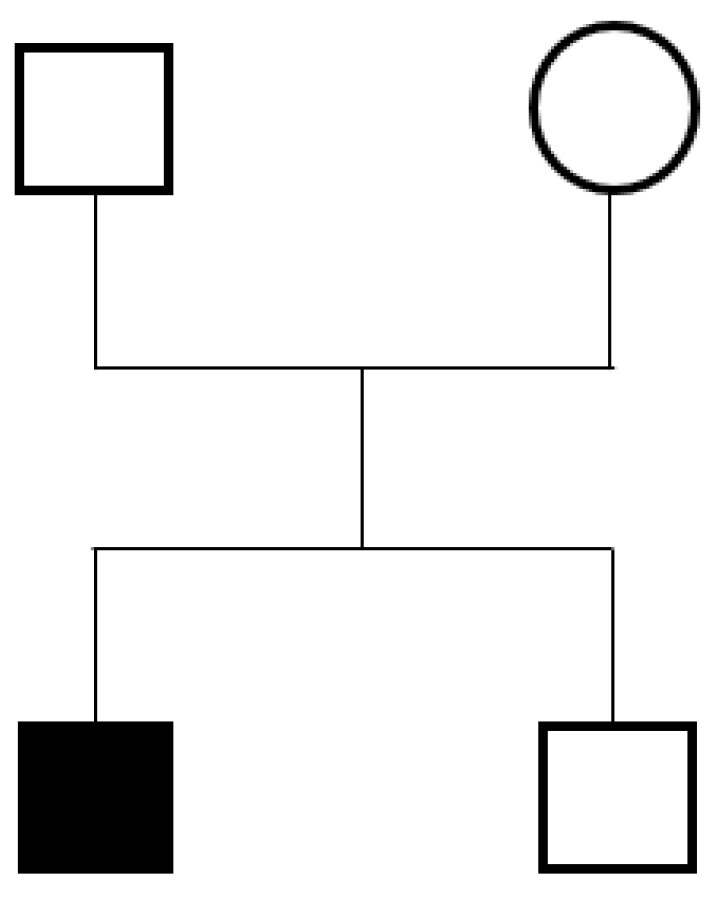
Pedigree of the index case.

**Figure 3 brainsci-12-01262-f003:**
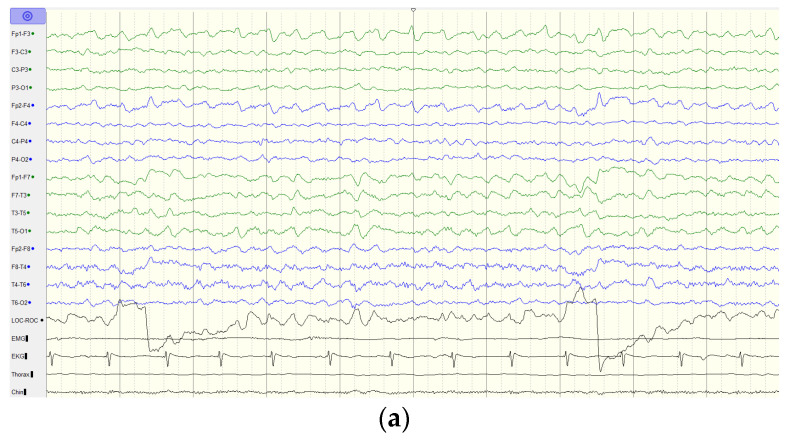
EEG findings of the index case. (**a**) EEG at the age of 11 years old shows generalized background slowing and continuous rhythmic 3 to 4 Hz slow waves over the left frontal, temporal, and occipital regions. (**b**) EEG at the age of 12 years old shows generalized background slowing and continuous polymorphic 4 to 5 Hz slow waves over the left hemisphere superimposed with focal spikes over T3. (**c**) EEG at the age of 13 years old shows rhythmic 3 to 4 Hz slow waves over the bilateral hemisphere. Green line indicates the left hemisphere and blue line indicates the right hemisphere.

**Table 1 brainsci-12-01262-t001:** Demographic data, clinical manifestations, neuroimaging findings, and clinical outcomes of 15 individuals with *UBTF* c.628 G>A (p.Glu210Lys) pathogenic variant-related neurodegeneration in childhood.

*UBTF* c.628 G>A (p.Glu210Lys) Pathogenic Variant-Related Neurodegeneration in Childhood	Total Number, *N* = 15
Gender, male: female	6:9
Clinical manifestation	
Developmental delay noted at ≤2 years, *n* (%)	10 (67)
Age at onset of neuroregression, years, median (range)	3 (2–7)
Initial motor regression, *n* (%)	6 (40)
Initial speech regression, *n* (%)	2 (13)
Initial motor and speech regression, *n* (%)	7 (47)
Microcephaly	8 (53)
Age at onset of epilepsy, years, median (range)	10.5 (5–15)
Neurologic examination	
Spasticity, *n* (%)	14 (93)
Dystonia, *n* (%)	11 (73)
Chorea, *n* (%)	3 (20)
Parkinsonism, *n* (%)	3 (20)
Ataxia, *n* (%)	10 (67)
Brain magnetic resonance imaging	
Supratentorial cerebral atrophy	15 (100)
Cerebellar atrophy	11 (73)
Diffuse white matter T2 hyperintensity	12 (80)
Thalamus involvement	2 (13)
Clinical outcome	
Profound intellectual disability	15 (100)
Non-verbal	13 (87)
Non-ambulatory	12 (80)

## Data Availability

Not applicable.

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
