# Peer review of "Clinico-Radiological Phenotype of UBTF c.628G>A Pathogenic Variant-Related Neurodegeneration in Childhood: A Case Report and Literature Review"

_brainsci, 2022, doi:10.3390/brainsci12091262_

Round 1
Reviewer 1 Report
None
Reviewer 2 Report
Chi et al present the clinical history of a new proband with UBTF-related neurodegenerative disease and review other patients in the literature to elucidate patterns in both clinical presentation and neuroradiological findings. Previous case reports and reviews were appropriately cited. The paper is very well written. My comments are quite minor. First, I wonder if the authors could comment as to why they pursued genome and not exome? Is that hospital protocol or would this variant somehow be missed on exome? Were any other variants of interest detected and was mitochondrial DNA included given the thalamic involvement? Next, supplemental table 1 is outstanding. I wonder whether a briefer table could be included in the main text to summarize in table form the salient features with the associated numbers/percentages detailed in the manuscript? Sometimes when written in prose the number get lost and can be re-emphasized in a table.
Reviewer 3 Report
My suggestions:
1. Was the patient scanned for another type of spasticity-related genes (such as SPG11, SPAST, etc). A table in a supplementary file would be interesting. If no, in the results you may mention, that no additional spasticity-related genes were found.
2. A figure on the family tree would also be interesting, even though no affected family members were found. You may also mention that family members refused the genetic test.
3. In the Results section, you may add a table, which summarizes all cases with UBTF E210K mutation.
4. Were any of the previous cases performed structure predictions on UBTF E210K mutation? If yes, you may mention the potential impact on the mutation. If no, a simple structure prediction would improve the paper further.
5. Were there any additional possibly pathogenic mutations described in the UBTF gene? You may mention it in the discussion briefly.
Round 2
Reviewer 3 Report
Manuscript can be accepted now